# Photoluminescent Microbit Inscripion Inside Dielectric Crystals by Ultrashort Laser Pulses for Archival Applications

**DOI:** 10.3390/mi14071300

**Published:** 2023-06-24

**Authors:** Sergey Kudryashov, Pavel Danilov, Nikita Smirnov, Evgeny Kuzmin, Alexey Rupasov, Roman Khmelnitsky, George Krasin, Irina Mushkarina, Alexey Gorevoy

**Affiliations:** Lebedev Physical Institute, 119991 Moscow, Russia; danilovpa@lebedev.ru (P.D.); smirnovna@lebedev.ru (N.S.); kuzmine@lebedev.ru (E.K.); rupasovan@lebedev.ru (A.R.); khmelnitskyra@lebedev.ru (R.K.); krasingk@lebedev.ru (G.K.); i.mushkarina@lebedev.ru (I.M.); a.gorevoy@lebedev.ru (A.G.)

**Keywords:** fluorides, diamond, ultrashort-pulse laser, direct laser inscription, photoluminescent microbits, vacancy clusters

## Abstract

Inscription of embedded photoluminescent microbits inside micromechanically positioned bulk natural diamond, LiF and CaF_2_ crystals was performed in sub-filamentation (geometrical focusing) regime by 525 nm 0.2 ps laser pulses focused by 0.65 NA micro-objective as a function of pulse energy, exposure and inter-layer separation. The resulting microbits were visualized by 3D-scanning confocal Raman/photoluminescence microscopy as conglomerates of photo-induced quasi-molecular color centers and tested regarding their spatial resolution and thermal stability via high-temperature annealing. Minimal lateral and longitudinal microbit separations, enabling their robust optical read-out through micromechanical positioning, were measured in the most promising crystalline material, LiF, as 1.5 and 13 microns, respectively, to be improved regarding information storage capacity by more elaborate focusing systems. These findings pave a way to novel optomechanical memory storage platforms, utilizing ultrashort-pulse laser inscription of photoluminescent microbits as carriers of archival memory.

## 1. Introduction

Photoluminescence (PL) is one of the most important optical processes, underlying relaxation of two-level quasi-molecular systems upon their complementary optical excitation [1]. Even single PL photons could be acquired and spatially resolved much easier than differential absorption of single photons. As a result, PL characterization in 2D- or 3D-scanning confocal micro- or nano-spectroscopy mode became an enabling tool for probing local molecular or crystalline structures [2,3], or electromagnetic near-fields [4,5].

Ultrashort-pulse lasers proved to work as a versatile tool for time-resolved and/or non-linear spectroscopy [6,7], precise surface nano- and micro-machining of any—absorbing or transparent—materials [8,9], micro-modification and inscription inside bulk transparent media [10,11,12]. In the latter case, (sub)microscale laser modification of molecular or crystalline structures and related PL spectra underlies facile and robust encoding of bulk diamonds for their tracing applications in identifying synthetic diamonds from natural ones in large commercial diamond collections [13], protecting trademarks of high-quality natural (potentially, synthetic too) diamond manufacturers [14], limiting commercial trading and marketing of illegal diamonds. This PL-based encoding appears unique to diamonds, where other popular encoding technologies—ablation fabrication of optically-contrasted (sub)microscale voids [15] or ablative birefringent nanogratings [16,17]—do not work in the ultra-hard diamond lattice, tending to be better for graphitization [18], while PL read-out is simpler and more sensitive. Similarly, many other crystalline scintillators and luminophores undergo ultrashort-pulse laser modification of their crystalline structures and related PL spectra [19,20,21], which is potentially promising for 3D optical encoding (writing/read-out) applications in storage devices (5D optical storage for specific advanced technologies [16]). Such 3D optical memory storage in bulk transparent media is an evergreen dream since the 1990s or even earlier [22], being highly promising and competitive compared to the previous 2D surface laser patterning compact disk (CD) and digital versatile (video) disk (DVD) technologies (common diameter—120 mm, thickness—1 mm, capacity—up to 17 GB for double-side, double-layer disks) utilizing 650 nm writing lasers, and the present 405 nm laser Blu-ray disk technology, supporting storage capacity up to 128 GB per four-layer disk. Other modern storage technologies enable even higher capacities—up to several TB, while possessing other technical advantages and drawbacks. Hence, rewritable, ultrahigh-capacity, low-power or autonomous high-speed memory, remaining robust to radiation, humidity and thermal shocks is still needed for quick-access and archival applications.

In this work we report a brief experimental evaluation study of natural diamond, LiF and CaF_2_ crystals as optical platforms for microscale photoluminescent encoding by ultrashort-pulse lasers for micromechanically-accessed archival optical storage, of their mechanisms of laser-induced color-center inscription, tests of potential 3D memory capacity and thermal stability.

## 2. Materials and Methods

In these studies, a 2 mm thick colorless brick of IaA-type natural diamond (total concentration of nitrogen atoms ≈ 130 ppm), and 5 mm thick slabs of undoped LiF and CaF_2_ crystals grown by Bridgman–Stockbarger method were utilized, being optically transparent at the writing laser wavelength of 525 nm (Figure 1a). The samples were characterized in the spectral range of 350–750 nm by room-temperature (RT) optical transmission microspectroscopy (Figure 1b), using an ultraviolet (UV)−near-IR microscope-spectrometer MFUK (LOMO, Saint-Petersburg, Russia). Inscription inside these bulk crystals at the depths of 100 μm (fluorides) and of 120 μm (diamond) in their transparency spectral regions (Figure 1b), accounting for their 525 nm refractive indexes of 2.4 (diamond), 1.4 (LiF) and 1.4 (CaF_2_) [23], was performed by second-harmonic (525 nm) pulses of the TEMA Yb-crystal laser (Avesta Project, Moscow, Russia) with the pulsewidth (full width at a half maximum) of 0.2 ps, repetition rate of 80 MHz split in pulse bunches of 0.05 ÷ 10 s duration by a mechanical shutter, and 50 nJ (average power—4 W) maximum output pulse energy E in the TEM_00_ mode. The 525 nm laser pulses with variable energies E up to 50 nJ were focused in a sub-filamentary regime (the 515 nm, 0.3 ps laser filamentation threshold energy ≈300 nJ (diamond) and 260 nJ (CaF_2_) [24]) by a 0.65 NA micro-objective into ≈1 μm wide spots (1/e-intensity diameter) inside the crystals, providing the peak laser fluence <6 J/cm^2^ and peak laser intensity <30 TW/cm^2^. The samples were mounted on a computer-driven three-dimensional motorized micropositioning stage and exposed in separate positions with variable transverse spacings in the range of 1–5 microns and longitudinal spacings in the range of 1–28 microns.

Since different alkali or alkali-earth fluorides are rather similar during high-temperature annealing due to high vacancy mobility (activation energy for diffusion ~0.1 eV [25]) and low-temperature aggregation [26], annealing of fluoride samples was performed only for the LiF sample at different temperatures in the range of 25–300 °C (20 min temperature ramp, 30 min stationary heating), using a temperature-controlled mount for Raman micro-spectroscopy, while the diamond sample was annealed in an evacuated oven for 1 h at different temperatures in the range of 25–1200 °C.

In our characterization studies, top-view and side-view (cross-sectional) photoluminescence imaging at the 532 nm continuous-wave pump laser wavelength and 100× magnification (NA = 1.45, spatial resolution ~1 μm) was performed by means of a Confotec MR520 3D-scanning confocal photoluminescence/Raman microscope (SOL Instruments, Minsk, Belarus) to measure relative intensity, spatial dimensions and optical PL-acquired separation of PL microbits (Figure 2).

## 3. Experimental Results and Discussion

### 3.1. Inscription of Photo-Luminescent Microbits

PL microbits were inscribed in the bulk crystalline CaF_2_ and LiF slabs, as well as in the diamond plate, at different pulse energies (Figure 2a–c, left side). Specifically, the PL microbits inside the CaF_2_ slab exhibit the corresponding energy-dependent microbit dimensions above the inscription threshold value of ≈3 nJ at the exposure of 10^7−^–10^9^ pulses/microbit. Similarly, the PL microbits were inscribed inside the LiF slab in the same energy range, while the threshold energy appears considerably higher (≈5 nJ, Figure 2b, left side) at the exposure of 10^7−^–10^9^ pulses/microbit, reflecting the higher bandgap energy of 13.0–14.2 eV in LiF [27,28], comparing to CaF_2_ with 11.5–11.8 eV [29,30]. Finally, in the diamond plate, the PL microbits were inscribed as a function of laser pulse energy, demonstrating their increasing dimensions at the exposure of 10^7−^–10^9^ pulses/microbit (Figure 2c, left side). Surprisingly, contrary to our expectations, the inscribed microbits appear inhomogeneous at higher magnifications (Figure 2a–c, right side) because of the non-linear photoexcitation/damage character and well-known high degree of clustering—up to nanoscale—for fluorine atoms to form dislocation loops [26] or around dislocations in the fluoride crystals [25]. In diamonds, such segregation of vacancies and interstitials also occurs in the form of multi-vacancies (voids) [31] or interstitial aggregates in B2-centers [31]. In the same line, these microbits look diffuse owing to the low diffusion energies of ~0.1 eV of interstitials and vacancies [26], facilitating room-temperature internal segregation and external collateral spreading of point-defect concentrations in the PL microbits.

In terms of spatial resolution of the PL microbits during the laser inscription process, even at low above-threshold pulse energies these features could be more or less resolved only at their 2 μm separation (Figure 2a–c, right side). The main reason for such moderate lateral (transverse) resolution is apparently the initial focal 1/e-diameter of 1 μm at the focusing NA = 0.65 (see Section 2—Materials and Methods), additionally increased by ≈1 μm lateral diffusion length of fs-laser generated electron-hole plasma during its electron-lattice thermalization over 1–2 picoseconds [32]. The corresponding point beam stability upon the focusing could result in negligible lateral displacements of ~10 nm. As a result, distinct resolution of the neighboring PL microbits becomes possible for their lateral separations, exceeding 2 μm distance. Meanwhile, it could be considerably improved till ~1–1.5 μm, utilizing specially designed high-NA (0.75–0.9) air focusing micro-objectives. Below, in Section 3.4, the longitudinal (interlayer) spatial resolution will be tested in the case of a brightly luminescent LiF crystal to evaluate the potential optical storage capacity of PL microbit arrays.

### 3.2. Photo-Luminescence Spectra of Microbits: Atomistic Inscription and Annealing Mechanisms

Typical PL spectra acquired in the laser-inscribed microbits by 3D-scanning confocal PL micro-spectroscopy are presented in Figure 3 in comparison to the corresponding spectra of the background non-modified materials. Specifically, the CaF_2_ slab exhibits the strongly enhanced PL yield in the region of 650–850 nm, peaked at 740 nm (Figure 3a). Though PL spectra of electronic excitations in fluorides are rather flexible due to high mobility of Frenkel defects and the multitude of their complexes [25,33,34], the observed peak could be assigned to some of these vacancy aggregates (F_x_^0,+^, where F is the fluorine vacancy with the trapped electron, x > 2 and upper indexes “^0,+^” denote the charged states) [25,26]. Similarly, in the LiF slab, the increased PL band in the range of 550–750 nm could be assigned to F_2_ (peak at 670 nm [33]) and F_3_ (peak at 650 nm [33]) centers, while the emerging PL band with its peak at 800–850 nm could also related to some as yet unknown F_x_^0,+^ centers [25,33,34].

Finally, the observed, strongly—by one order of magnitude—enhanced PL band in the micromark inscribed inside the diamond slab exhibits the main spectral features, representing the neutral (NV^0^, zero-phonon line, ZPL, at 575 nm [31]) and negatively charged nitrogen-vacancy (NV^−^, zero-phonon line at 637 nm [31]) centers of substitutional nitrogen atoms with a photo-generated vacancy, as well as their red-shifted phonon replica.

Atomistic processes underlying the observed laser-induced transformations of PL spectra in LiF and CaF_2_ are supposed to be associated with aggregation of mobile neutral (I-center [25,26]) and negatively charged (F-center [25,26]) vacancies, along with fluorine neutral (H-center [25,26]) and negatively charged (α-centers [25,26]) interstitials (Equation (1)), approaching to hundreds of aggregated defects usually concentrated in dislocation loops [25,26]. Likewise, in the diamond plate, the mobile photo-generated vacancies could be trapped by substitutional nitrogen atoms (C-centers [31]), resulting in well-known neutral or charged NV complexes [18,31] (Equation (1)):$$florides:H+H\to H_{2},H+V_{K}\to F_{3}^{-},$$
(1)$$diamond:N_{S}+V^{0}\to NV.$$

Equation (1)—Atomistic processes, resulting in photo-induced vacancy complexes in fluorides and diamond.

In the same line, one can see strong stationary annealing of mobile vacancy-related color centers in LiF already at temperatures elevated by 200–300 °C (Figure 4a), almost deleting the microbit signal. In contrast, in denser and more rigid diamond lattice the Frenkel vacancies anchored by C-centers, remain rather stable even at high temperatures, approaching 1200 °C (Figure 4b).

### 3.3. Photogeneration of Frenkel Pairs of Point Defects in LiF during Atomistic Inscription

PL yield at 670 nm—in the peak related to F_2_-centers—was used to track fs-laser photogeneration of Frenkel pairs in LiF, underlying the formation of these centers. As can be seen in Figure 5a,c, the PL yield in LiF exhibits the non-linear (power slope in the range of ≈3.3–3.8) monotonic dependence on pulse energy E = 2.5–13 nJ (peak fluence ≈ 0.3–1.7 J/cm^2^, peak intensity ≈ 1.5–9 TW/cm^2^) (previously—in diamond [35]) and exposure of (4–800) × 10^6^ pulses/spot (at room temperature, Figure 5b,d). Moreover, the abovementioned annealing effect at the temperatures of 200 °C and 300 °C results not only in the decreased PL intensity at 670 nm (Figure 5c), but also in the different exposure trends (Figure 5d) apparently related to cumulative heating of the material at the ultra-high 80 MHz exposure of the static sample, which is well-known to be favorable for self-trapped exciton stabilization via Frenkel pair formation [25]. The cumulative heating effect is more pronounced at room temperature (Figure 5d), while the elevated temperatures make it less distinct.

We have analyzed the observed PL yield at 670 nm vs. pulse energy E in LiF (Figure 5c), representing the concentration of F_2_-centers in the probed confocal volume, alike to our previous similar studies of NV-center yield upon fs-laser exposure in diamond [35]. According to high bandgap energy of E_dir_(Γ-point) ≈ 13.0–14.2 eV in LiF [27,28], formation of F_2_-centers requires either N = E_dir_/ħω ≈ 6 photons at the 525 nm wavelength (photon energy ħω ≈ 2.4 eV), or “hot” non-equilibrium electron of this energy (effectively, considerably higher to fulfill both the quasi-momentum and energy conservation laws). The evaluated laser-induced prompt ponderomotive enhancement of the bandgap [36,37], U_p_ = e^2^E^2^/(4m_opt_ω^2^), is minor (<1 eV) in the utilized intensity range of 9–30 TW/cm^2^ (electric field strength E = 15–30 MV/cm) for the arbitrary optical mass of electron-hole pair m_opt_ = m_e_m_h_/(m_e_+m_h_) = m_0_/2, assuming m_e_,m_h_ = m_0_ (free-electron mass).

Recently, for such analysis of electron-hole plasma and PL dynamics, a kinetic rate model for electron-hole plasma density ρ_eh_ was enlighteningly used in the common form [38], including (1) ultrafast, pulsewidth-limited multiphoton (cross-section σ_N_), (2) impact ionization (coefficient α), (3,4) fast picosecond three-body Auger (coefficient γ) and slow binary radiative (coefficient β) recombination, as well as (5) fast self-trapping of electron-hole pairs (excitons, characteristic time τ_str_) to produce one color center per self-trapped excitons [25] as the consecutive terms, respectively, and describing the corresponding continuous-wave-laser pumped PL yield Φ_PL_ as follows (Equation (2)) [32]:(2)$$\frac{d\rho_{eh}}{dt}=\sigma_{N}I^{N}+\alpha I\rho_{eh}-\gamma \rho_{eh}^{3}-\beta \rho_{eh}^{2}-\frac{\rho_{eh}^{2}}{\tau_{str}},\Phi \propto \int \rho_{eh}^{2}dt$$
(3)$$\frac{d\rho_{eh}}{dt}=\sigma_{6}I^{6}-\gamma \rho_{eh}^{3}-\frac{\rho_{eh}^{2}}{\tau_{str}},\sigma_{6}I^{6}\approx \gamma \rho_{eh}^{3},\rho_{eh}\propto I_{0}^{2},\Phi \propto \int \rho_{eh}^{2}dt\propto I_{0}^{4}.$$

Equations (2) and (3)—Kinetic rate equations for electron-hole plasma and related PL yield Φ_PL_ of the F_2_-centers produced via exciton self-trapping [25]: (2) general form, (3) case-specific form for our experiments in LiF.

In the case of LiF, Equation (2) could be presented in the case-specific form (Equation (3)), where only six-photon ionization and Auger recombination balance each other, while excitonic self-trapping accompanies the electron-hole plasma relaxation. As a result, PL yield could follow the non-linear dependence on the peak fs-laser intensity I_0_ with the power slope ≈4, being consistent with the measured values of 3.3–3.8 (Figure 5c). For the used nJ-level pulse laser energies, strong electron-hole plasma absorption is not achieved [32,39], thus enabling rather delicate inscription of PL nano- and microbits.

For comparison, in diamond, both the energy and exposure dependences of the NV^0^ and NV^−^ color center yield exhibit non-linear (Figure 6a,c) and linear (Figure 6b,d) trends, respectively. Only linear dependence of the PL intensity on exposure time indicated the proceeding, unsaturated accumulation of the color centers. However, in terms of the pulse energy, PL intensity of NV^−^ centers exhibit highly-nonlinear yield (power slope—5.5 ± 0.2), while the corresponding weaker NV^0^ peak (Figure 6a,b) rises similarly above the noise level at higher pulse energies (Figure 6c).

Similarly to Equations (2) and (3), in the case of diamond, its weak non-linear photoexcitation process across the minimal direct Γ-point bandgap of 7.3 eV, which is distinct in Figure 6c, could be represented in the following familiar form [32]
(4)$$\frac{d\rho_{eh}}{dt}=\sigma_{3}I^{3},\;\;\rho_{eh}\propto I_{0}^{3},\;\;\Phi \propto \int \rho_{eh}^{2}dt\propto I_{0}^{6}.$$

Equation (4)—Kinetic rate equations for three-photon excitation of electron-hole plasma and related PL yield Φ_PL_ produced via exciton self-trapping in diamond [31].

Here, marginal photogenerated electron-hole pairs become intermixed in the EHP, losing their initial correlation during the photogeneration and appear independently with the overall 2N-fold slope in the excitonic recombination [32], preceding NV-center formation. The observed difference in the fs-laser driven formation of Frenkel I–V pairs in LiF and diamond is apparently related to their drastically differing ionicity, favorable for excitonic self-trapping in fluorides [25,26], as compared to the predominating electron(hole)-lattice interactions in diamond [32].

### 3.4. Evaluation of Storage Capacity Utilizing Photo-Luminescent Microbit Arrays

Finally, we have performed experimental evaluation of PL microbit density, which is the key characteristic of archival optical storage. Above, we inscribed PL bits in the natural diamond, LIF and CaF_2_ samples with ≥2 μm lateral separation, exceeding the micropositioning accuracy, which could be easily resolved in the PL images (Figure 2). Furthermore, we undertook inscription and confocal PL visualization of separate linear arrays of PL microbits with variable vertical (depth) separation changed in the range of 1–28 μm (Figure 7), in order to evaluate the minimal resolvable vertical separation. PL visualization was performed by means of Olympus (40×) and Nikon (100×) microscope objectives with vertical resolution Δz = 1 or 2 μm, respectively (Figure 7a,b). The corresponding side-view PL imaging results for the same microstructure of paired linear arrays are presented in Figure 7c,d.

Here, one can find that the pairs of microbit lines become visibly separable, starting from 11 μm for the 1 μm resolution visualization (Figure 7c), while only at 16 or 21 μm—for the 2 μm resolution visualization (Figure 7d). Hence, accounting for the appropriately resolvable 2 μm intra-layer separation of microbits and their 11 μm inter-layer separation, one can evaluate the bulk microbit density of 25 Gbits per cubic centimeter for the simple cubic lattice of microbits, i.e., about 3 Tbits per disk for the 120 mm diameter of the standard CD or DVD disks and the 10 mm thickness. This optical storage capacity is comparable to previous optical memory writing technologies (visible microvoids [15], birefringent nanotrenches [16,17], photoluminescent microbits [40]), but they have clear benefits in the confocal non-linear memory read-out due to non-destructive laser inscription technology. Moreover, considerable, few-fold additional increase in the storage capacity could be achieved by higher-NA (NA > 0.65) inscription and other advanced optical means.

## 4. Conclusions

In this study, bulk high-NA inscription (writing) of photo-luminescent microbits in LiF, CaF_2_ and diamond crystals, as a delicate laser micromachining process, was performed by means of ultrashort-pulse laser and tested (read-out) by 3D-scanning confocal photoluminescence micro-spectroscopy. Preliminarily, optical storage density for the simple cubic lattice of microbits was evaluated as high as 25 Gbits/cm^3^, provided by the precise micromechanical positioning, but could be few-fold increased by using more sophisticated optical focusing tools during the encoding procedure. The underlying photo-luminescent color centers were identified in the fluorides (fluorine vacancy-based F_2_-centers and similar vacancy-based specifies) and diamond (carbon vacancy-based NV-centers) by PL micro-spectroscopy, while their laser inscription mechanism was revealed in fluorides for the first time, comparing to the more or less known mechanisms for synthetic and natural diamonds. Moreover, the color centers could be easily annealed in the fluorides at moderate temperatures of 300 °C due to the high lability of the centers and room-temperature mobility of their atomistic constituents, comparing to the relatively robust color (NV) centers in diamond, persisting even at rather elevated temperatures of ≈1200 °C. Our first-step research highlights the way for potential implications of laser-inscribed photo-luminescent microbits in archival optical storage with micromechanical access for its read-out.

## Figures and Tables

**Figure 1 micromachines-14-01300-f001:**
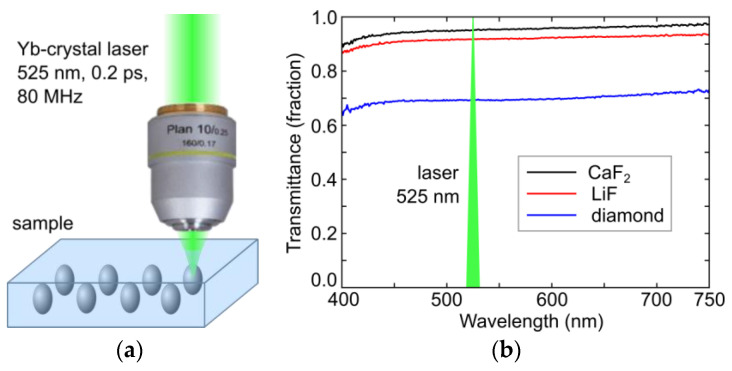
(**a**) Laser inscription setup, sketching the bulk PL-microbit inscription procedure; (**b**) transmittance spectra of natural diamond, LiF and CaF_2_ crystals, with the laser writing wavelength of 525 nm shown in their transparency spectral region by the green triangle.

**Figure 2 micromachines-14-01300-f002:**
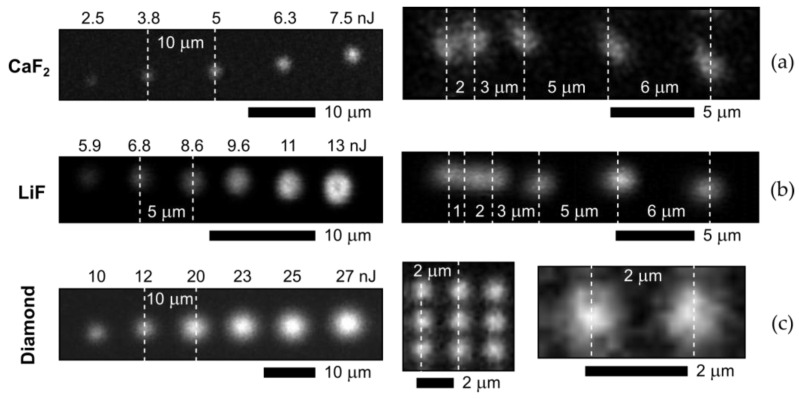
(**left**) Top-view PL images of linear slices of square PL microbit arrays inscribed at different laser conditions and acquired at 755 nm in CaF_2_ (**a**), at 650 nm in LiF (**b**), at 650 nm in natural diamond (**c**). (**right**) Their corresponding front-view images of neighboring PL microbits inscribed in these dielectrics at different lateral separations, varying in the range of 1–6 μm.

**Figure 3 micromachines-14-01300-f003:**
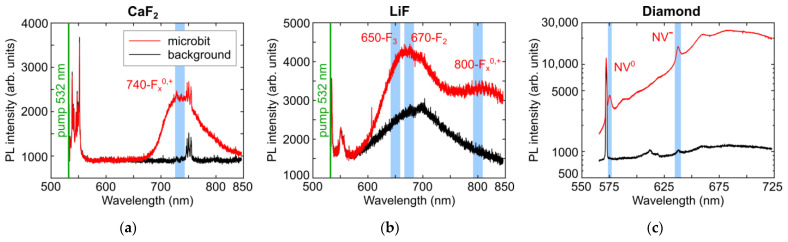
PL spectra of separate PL microbits (red curves) inscribed in CaF_2_ (**a**), LiF (**b**) and natural diamond (**c**) regarding their background spectra of the unmodified materials (dark curves).

**Figure 4 micromachines-14-01300-f004:**
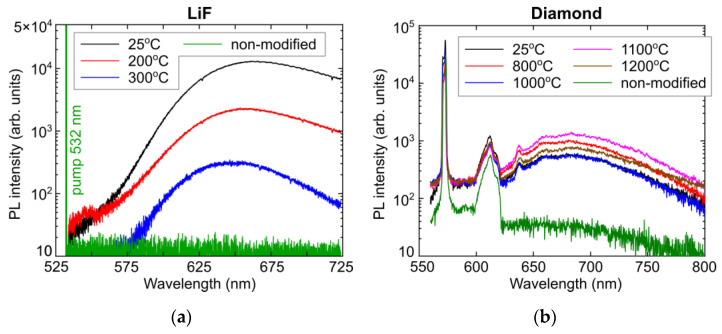
PL spectra of LiF (**a**) and natural diamond (**b**) upon annealing in the corresponding different temperature ranges, regarding the unannealed non-modified materials.

**Figure 5 micromachines-14-01300-f005:**
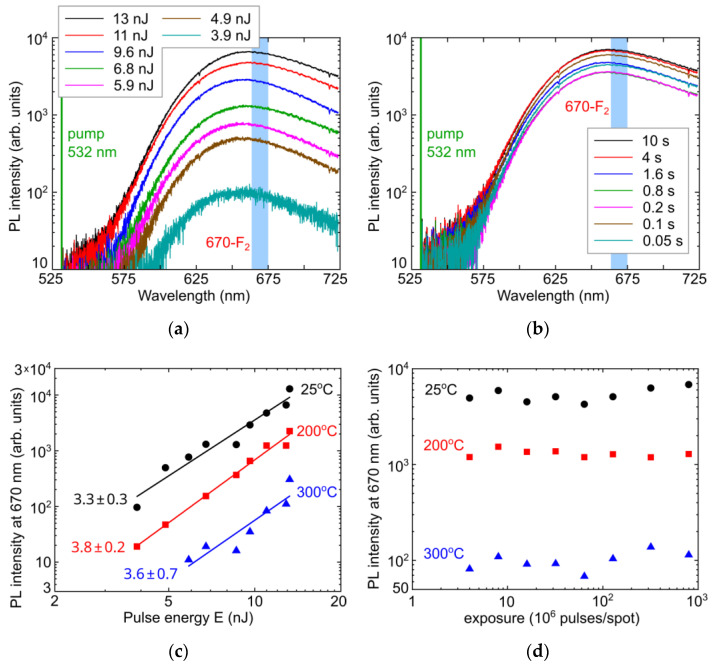
(**a**) PL spectra of microbits in LiF inscribed at variable pulse energy (see the frame inset) and the fixed exposure of 10 s (×80 MHz), spectral assignment after [33]; (**b**) PL spectra of microbits in LiF inscribed at the variable exposures (see the frame inset) and the fixed pulse energy of 13 nJ (spectral assignment after [33]); (**c**) PL intensity of 670 nm (F_2_-center [33]) peak in the spectra as a function of pulse energy (peak fluence—0.2–2.4 J/cm^2^, peak intensity—1–12 TW/cm^2^) at the maximal exposure of 10 s without annealing (25 °C, black circles) and after annealing at 200 °C (red squares) and 300 °C (blue triangles) as well as their linear fitting curves of the same colors with the corresponding slopes; (**d**) PL intensity of 670 nm (F_2_-center [33]) peak in the spectra as a function of exposure at the pulse energy of 13 nJ without annealing (25 °C, black circles) and after annealing at 200 °C (red squares) and 300 °C (blue triangles).

**Figure 6 micromachines-14-01300-f006:**
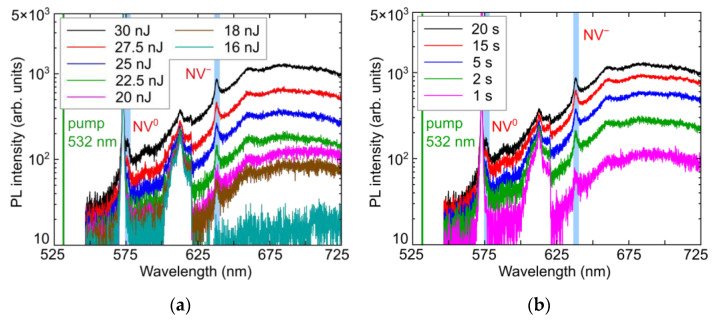
(**a**) PL spectra of microbits in diamond inscribed at variable pulse energy (see the frame inset) and fixed exposure of 20 s (×80 MHz), spectral assignment after [31]; (**b**) PL spectra of microbits in diamond inscribed at variable exposure (see the frame inset) and fixed pulse energy of 30 nJ (spectral assignment after [31]); (**c**) PL intensity of NV^0^ (575 nm [31], black circles) and NV^−^ (637 nm [31], red squares) peaks in the spectra as a function of pulse energy at exposure of 20 s and linear fitting curve for NV^−^ with its slope indicated; (**d**) PL intensity of NV^0^ (black circles) and NV^−^ (red squares) peaks in the spectra as a function of exposure at energy of 30 nJ.

**Figure 7 micromachines-14-01300-f007:**
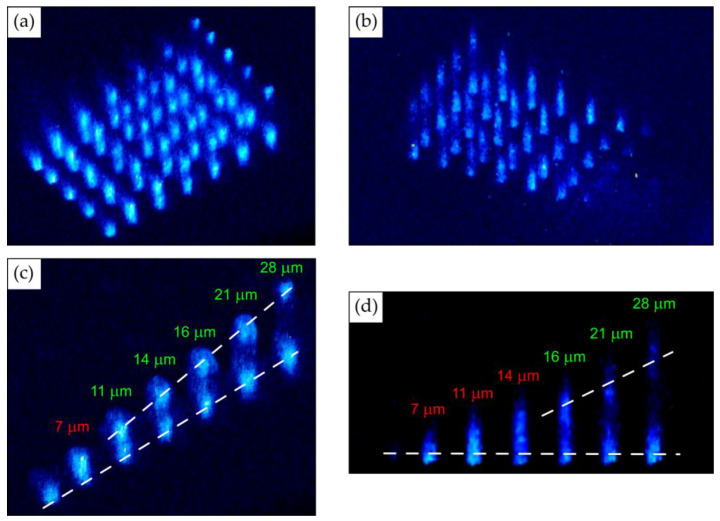
PL imaging of stair-like set of pairs of linear microbit arrays in LiF with variable intra-pair vertical separation in the range of 1–28 μm using Olympus (**a**,**c**) and Nikon (**b**,**d**) microscope objectives with the vertical resolution Δz = 1 or 2 μm, respectively: (**a**,**b**) 3D view; (**c**,**d**) side-view images of neighboring PL microbit arrays, showing their resolvable separation, starting from 11 μm (**c**) or 16 μm (**d**).

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
