# Peer review of "Photoluminescent Microbit Inscripion Inside Dielectric Crystals by Ultrashort Laser Pulses for Archival Applications"

_micromachines, 2023, doi:10.3390/mi14071300_

Round 1

Reviewer 1 Report

The manuscript entitled “micromachines-2444228” dealing with the ultrashort laser pulses has been reviewed. The paper has been nicely written but needs some improvement. Please follow my comments.

1.     What is the industrial application of ultrashort laser pulses?

2.     Please proofread the paper.

3.     What is the main research question for this research work?

4.     Conclusion is too short and is not following the main part of the paper.

5.     What is the future direction of this work? Please add it in the segmented section.

6.     The absorptivity of laser in drives many factors such as penetration. Refer to the following paper to highlight this note in your manuscript. “The effect of absorption ratio on meltpool features in laser-based powder bed fusion of IN718”.

English needs more care.

Reviewer 2 Report

The authors fabricated photoluminescent microbits in CaF2, LiF, and natural diamond with a femtosecond laser at 525 nm wavelength. The photoluminescence spectrum of those obtained microbits was investigated, and the characteristic PL peaks were identified and recognized to certain F/N atomic states. The effect of laser energy, pulse number, and annealing post-process has been studied. By checking the spatial separation of microbits, an impressive microbit density could be achieved in LiF, which demonstrates its huge application potential in the information storage field. Overall, the results are interesting to the broad readership of Micromachines. However, there are lots of important discussions and information missing. Before this manuscript becomes acceptable, the authors are required to address the following comments well.

1, In the introduction part, the authors claimed that they conducted an evaluation study of natural diamond, LiF, and CaF2 crystals. However, in the manuscript, the evaluation of the annealing effect is only reported for LiF and diamond while only the results on LiF showing the effect of laser parameters and spatial resolution test are included in the manuscript. Please add the missed parts or clarify why selectively show the results.

2. Please add the scale bar in Figure 2.

3, In Figure 3b, what does the peak at 700 nm represent?

4, Please add descriptions of the symbols used in equation 1. And keep the font format consistent among the equations.

5, The discussion in Figure 5d should be strengthened. The trends of 25, 200, and 300 degrees are quite different. Why does the 200-degree case remain stable while the 300-degree case first decreases and then increases? In addition, the data points from the 25-degree case are less than 200 and 300-degree cases. What’s the reason for that?

6, The sentence (“previously -in diamond [26]”) in line 168 is misleading. Please clarify the meaning of this sentence.

7, There are several typos in the manuscript:

Line 57-61: the value format of pulse duration, pulse energy and spot size;

Line 166: “Figs. 5a,c”;

Line 226: “Figs. 6c,d”;

Line 235: “staring from”

Based on the abovementioned comments, this manuscript is recommended for major revision. A revised manuscript is required. 

Reviewer 3 Report

The authors designed and presented their detailed experiments to achieve microbit inscription inside dielectric crystals with ultrashort laser pulses. The results are scientific and persuasive, and the idea of optical inscription should definitely benefit the readers. Therefore, we recommend that the paper get published, after some improvement in the introduction section. The present introduction is too short and cannot provide enough information in the literature review of the topic.

Round 2

Reviewer 2 Report

The authors have addressed the comments well. One more comment is further polish the figure to make the legend stay in the figure region.